# Epidemiological survey of anterior segment diseases in Japanese isolated island using a portable slit-lamp device in home-based cases in Miyako Island

Eisuke Shimizu[1,2,3]*, Kazuhiro Hisajima[4], Shintaro Nakayama[1,3], Hiroki Nishimura[1,2,3], Rohan Jeetendra Khemlani[1,2], Ryota Yokoiwa[1], Yusuke Shimizu[1], Masato Kishimoto[4], Keigo Yasukawa[4,5]

1 OUI Inc., Tokyo, Japan, 2 Yokohama Keiai Eye Clinic, Kanagawa, Japan, 3 Department of Ophthalmology, Keio University School of Medicine, Tokyo, Japan, 4 Dr. Gon Kamakura Clinic, Kanagawa, Japan, 5 Dr. Gon Clinic, Okinawa, Japan

* ophthalmolog1st.acek39@keio.jp

**Data Availability Statement:** There are ethical restrictions on sharing the minimal data for this study publicly, because they contain potentially

## Abstract

The ophthalmic diseases often affect the elderly and require proper diagnosis, treatments, and follow ups. However, many cases lack adequate eye care due to limited resources and decreased daily living activities among individuals. Despite the importance of ophthalmic home care, epidemiological research in this area has been lacking. This study utilized portable ophthalmological equipment to conduct an epidemiological research of anterior segment diseases in patients receiving home-based medical care in Japanese isolated island "Miyako island". A retrospective examination was conducted on home visit cases from a single facility in Miyako island (Dr. Gon Clinic). Data from 147 cases were collected and analyzed which the data are all recorded by the home care doctors and send the data to the cloud in order to make a diagnosis by the ophthalmologists. Findings included various anterior segment eye diseases such as Ptosis, meibomian gland dysfunction, conjunctival chalasis, corneal opacity, conjunctival hyperemia, pterygium, and cataract. Notably, over half of the cases (51.67%) had not undergone cataract surgery although the average age are notably high (85.69 ± 12.11 age of years). Among those without surgery, all showed signs of moderate to severe cataracts, with severity increasing with age. Additionally, a shallow anterior chamber depth was observed in one-third of these cases (33.63%) which considerable to receive a surgery to prevent the acute glaucoma attack. From our study, to ensure that patients in remote islands as Miyako island do not miss opportunities for eye care, there is an urgent need for the establishment of a supportive medical system.

## Introduction

Japan, a developed country, has a significantly higher ratio of ophthalmologists compared to the global average. However, there is a notable disparity in the provision of ophthalmic care between urban and rural areas, with a significant shortage of specialists in rural regions [1]. Ophthalmic diseases frequently affect the elderly population, necessitating proper diagnosis,

identifiable participant information. Data are available upon request from Yu Matsumoto, Keigankai Yokohama Keiai Eye Clinic information manager, via email (info@keigankai.com) for researchers who meet the criteria for access to confidential data.

**Funding:** This work was supported by the Japan Agency for Medical Research and Development, Uehara Memorial Foundation, Hitachi Global Foundation, Kondo Memorial Foundation, Eustylelab, Kowa Life Science Foundation, Keio University Global Research Institute, The Asahi Glass Foundation, The Okawa Foundation, Suzuken Memorial Foundation, Yakult bioscience Foundation and Daiwa Securities Foundation. No other funding statement to declare including company and patent. The funders had no role in study design, data collection and analysis, decision to publish, or preparation of the manuscript.

**Competing interests:** E.S. is a founder of OUI Inc. OUI Inc. has the patent for the Smart Eye Camera (Patent No. JP; 6627071, USA; 16/964822, EU; 19743494.7, China; 201980010174.7, India; 202017033428, VN; 1-2020-04893, and Africa; AP/P2020/012569. Patent pending EU; 2175926.2, US; 17/799043). There are no other relevant declarations relating to this patent. OUI Inc. did not have any role in the funding, study design, data collection and analysis, decision to publish, or preparation of the manuscript. The authors declare no competing interest associated with this manuscript.

treatment, and follow-up. However, many cases lack adequate eye care due to resource constraints and decreased activity levels among individuals, particularly in rural areas and developing regions such as Africa [2, 3]. The lack of robust epidemiological data on pediatric eye disorders in Ethiopia impedes the planning and evaluation of preventive and curative services for children with ocular morbidity [4]. Additionally, there is a critical need for epidemiological studies to investigate the risk factors for cataract development, which are essential for planning and implementing effective global preventive programs, especially in developing countries [5]. Global estimates and projections of vision impairment highlight major obstacles, including the lack of epidemiological data in developing countries, which hampers the effective delivery of eye care services [6]. Despite the importance of ophthalmic home care, epidemiological research in this area is limited due to the polarization of ophthalmological resources [7].

Historically, epidemiological studies on ophthalmic diseases have predominantly used population-based examinations, common in cohort studies. However, there has been a lack of focus on home-visit cases, particularly for patients who face challenges in attending hospital appointments [8].

This study aims to address this gap by conducting an extensive epidemiological survey. We screened home-visit patients from a single institution using portable ophthalmological devices [9–16], with diagnoses subsequently confirmed by ophthalmologists via telemedicine. The objective of this research is to screen the anterior segment of the eyes using a portable slit-lamp microscope in a cohort of patients receiving home care in the Miyako Island area and to investigate the prevalence and characteristics of ophthalmologic diseases in this cohort.

## Materials and methods

### Study design

This retrospective study was conducted in accordance with the Declaration of Helsinki and approved by the Ethics Review Committee of the Kobawa Medical Corporation (Committee No.: 21000056, No. 20230501). As all data were anonymized and collected retrospectively, written informed consent was not obtained; instead, an opt-out procedure was employed. The study involved analyzing anterior segment videos recorded using a portable slit-lamp microscope (Smart Eye Camera: SEC, OUI Inc., Tokyo, Japan) at a non-ophthalmic institution, Dr. Gong Clinic (Okinawa, Japan). These videos were then evaluated by ophthalmologists at Yokohama Keiai Clinic (Kanagawa, Japan). The inclusion criteria were: (1) all patients who received a visit by a doctor at Dr. Gong Clinic from November 2021 to March 2022, and (2) patients whose anterior segment was captured using the SEC on the same day. The exclusion criteria included: (1) cases where the patient had difficulty imaging the anterior segment, such as inability to open the eyelids or severe body movements, and (2) cases where the patient refused imaging due to dementia or insufficient understanding and cooperation. Based on these criteria, 147 cases (294 eyes) were selected for anterior segment video analysis by ophthalmologists remotely. We did access to the database from 2024 April 1st to May 31st in order to accessed for research purposes. Moreover, authors only accessed to de-identification data which data can specify individuals.

### Portable slit-lamp microscope

The SEC, a smartphone attachment medical device used as a portable slit-lamp microscope, was utilized in this study. Previous animal studies [9] and several clinical studies [10–16] have demonstrated the SEC's diagnostic capabilities comparable to conventional slit-lamp microscopes in Japanese, Indian, Indonesian, and Italian objects. The SEC can emit slit light at 0.1 to 0.3 mm when observing the lens in a mydriatic and non- mydriatic status [10, 11]. Additionally, the SEC

can record moving images of the anterior segment, allowing the collection of extensive video and image data. Image resolution is set at 720 × 1280 pixels to 1080 × 1920 pixels, with frame rates of 30 or 60 frames per second. An iPhone 7 (Apple Inc., Cupertino, CA, USA) was used for recording. Non-ophthalmologists at Dr. Gong Clinic performed a minimum of 10 seconds of anterior segment imaging for each eye using the SEC (Fig 1).

## Anterior segment assessments

Ophthalmologists evaluated the anterior segment videos recorded by the SEC using a SEC cloud-based image filing system (Fig 2). The anterior segment of the eyes was divided into seven regions: eyelid, eyelashes, conjunctiva, cornea, anterior chamber, iris, and lens, with each region evaluated separately. For the eyelids, the presence or absence of ptosis, chalasis, entropion, and meibomian gland dysfunction (MGD) were assessed. Eyelashes were evaluated for the presence or absence of entropion and trichiasis. The conjunctiva was examined for hyperemia, papillae, follicles, and chalasis. The cornea was assessed for opacity and keratic precipitates (KP). The anterior chamber was evaluated for depth, cells, and flare. The iris was assessed for pupil, damage or atrophy, and pseudoexfoliation (PE). For the lens, the presence of an intraocular lens (IOL) or cataracts or aphakia were evaluated using the Emery-Little classification (NS1-5) [17]. Multiple ophthalmologists (E.S. and Y.S.) evaluated all anterior segment videos.

## Data and statistical analysis

The cross-sectional study targeted homebound patients residing on Miyako Island. Given the retrospective nature of the study, defining a precise target sample size in advance was challenging. Consequently, data that met the inclusion criteria were aggregated over a defined period.

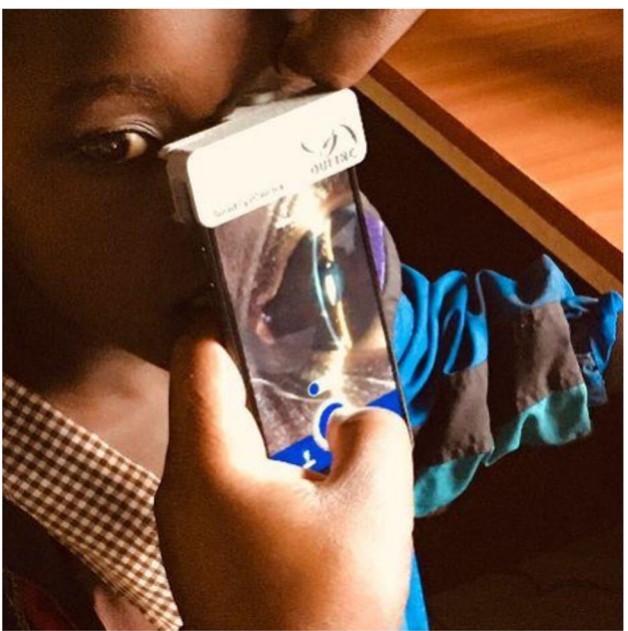
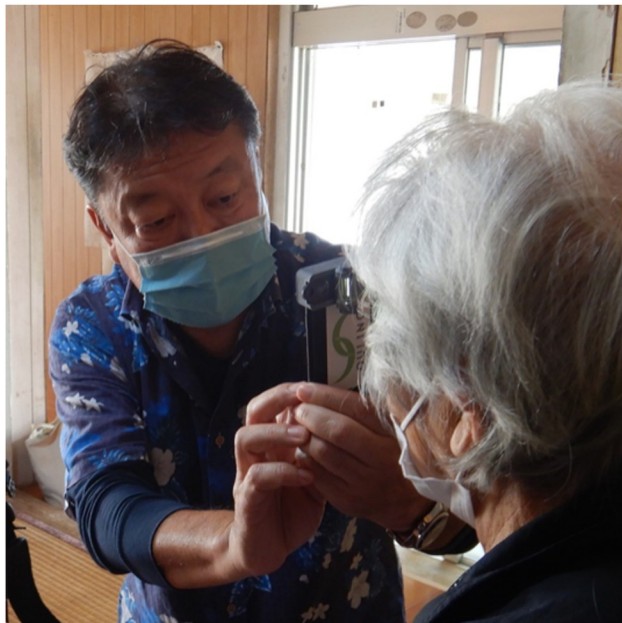

**Fig 1. Anterior segment imaging.** Anterior segment imaging captured using the Smart Eye Camera (left), showcasing the device's capability in delivering high-quality diagnostic images for ophthalmic evaluation. On the right, a co-author demonstrates the practical application of the Smart Eye Camera at the Dr. Gong Clinic, highlighting its portability and ease of use in a clinical setting for the assessment of anterior segment conditions.

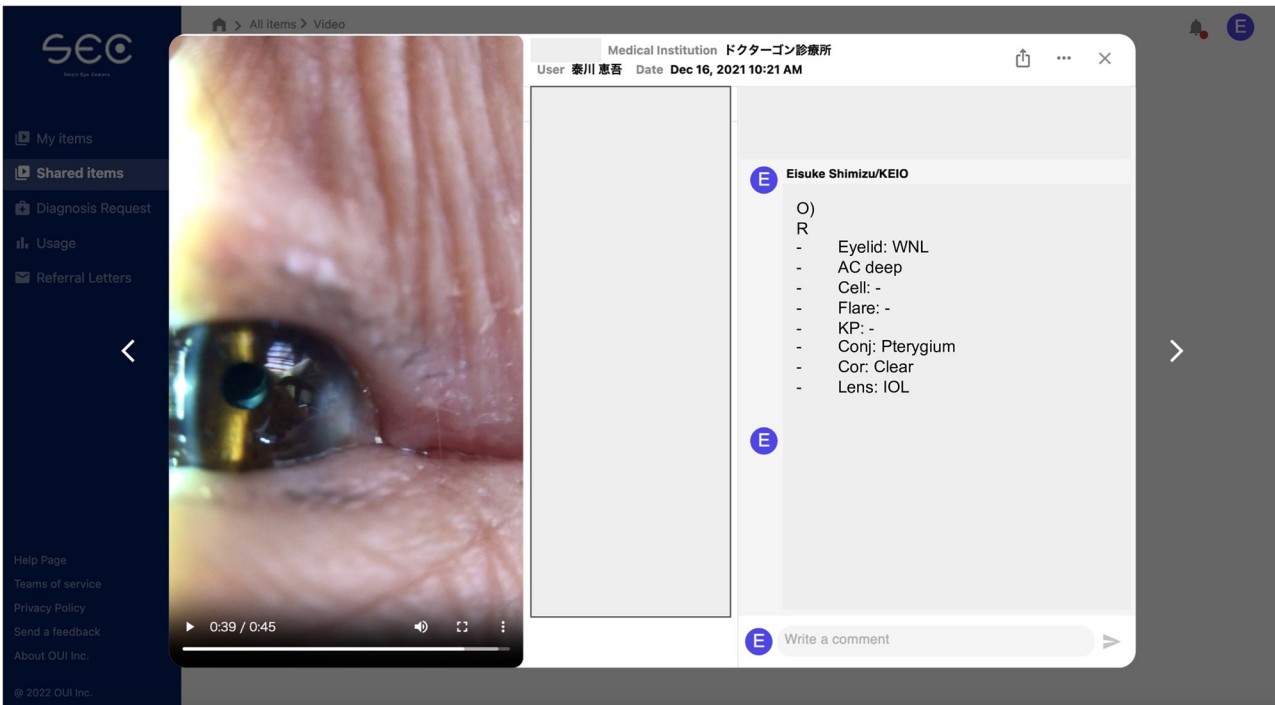

**Fig 2. Teleophthalmology assessment.** Online assessment of findings in the anterior segment of the eyes using the Smart Eye Camera (SEC) cloud-based image filing system. This system allows clinicians to upload and store high-resolution images securely, facilitating remote evaluations, consultations, and collaborative decision-making. The cloud-based platform enhances accessibility and ensures efficient management of patient data, enabling real-time monitoring and comprehensive assessments from various locations.

The prevalence of ophthalmologic conditions was calculated based on the number of observable eyes exhibiting specific findings. All analyses were conducted using Google Spreadsheets (Google LLC, Mountain View, CA, USA).

## Results

### Basic characteristics and prevalence of anterior segment diseases

Anterior segment videos were collected from 294 eyes of 147 patients. The mean age of the patients was 85.69 ± 12.11 years, with a male-to-female ratio of 55:92 (Fig 3). The most common eyelid abnormality was MGD, present in 83.76% of patients, followed by blepharochalsis (2.16%) and ptosis (1.08%). Abnormalities in the eyelashes included trichiasis in 0.35% of patients. Conjunctival abnormalities were observed as follows: conjunctival chalasis in 23.44%, hyperemia in 14.08%, follicles in 2.67%, and pterygium in 6.74% of cases. Corneal abnormalities included corneal opacity in 10.99% and band keratopathy in 1.06% of cases. Findings in the anterior chamber showed 84.36% of patients had a deep anterior chamber depth, while 15.64% had a shallow anterior chamber depth. Iris findings indicated that 97.83% had a normal pupil, 2.17% had an irregular pupil, and 1.08% had iris damage or atrophy. Regarding lens findings, 48.33% of patients had an intraocular lens (IOL), and no cases of aphakia were identified. All patients exhibited cataracts, with the following distribution: 0.74% in NS1, 22.30% in NS2, 20.82% in NS3, 5.95% in NS4, and 1.86% in NS5 (Fig 3). Other findings included eyelid papilloma (0.36%), lower eyelid nevus (0.36%), pinguecula (2.22%), cornea dermoid (0.41%), small pupil (0.73%), IOL donesis (1.48%), and IOL dislocation (0.74%) (Fig 4).

| Age | 85.69±12.11 |
|---|---|
| Gender | |
| Male | 55 |
| Female | 92 |

**Lens findings**

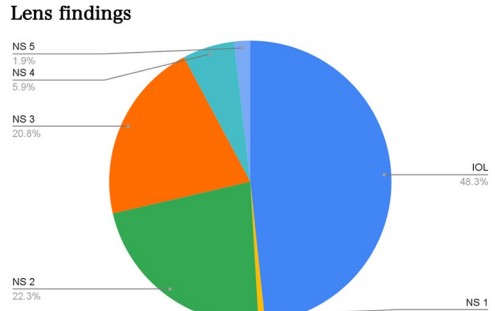

NS 5 1.9%
NS 4 5.9%
NS 3 20.8%
NS 2 22.3%
NS 1 0.7%
IOL 48.3%

MGD: meibomian gland dysfunction, KP: keratic precipitates, PE: pseudoexfoliation, IOL: intraocular lens, NS: nuclear sclerosis.

| | % | | | % |
|---|---|---|---|---|
| **Eyelids** | | **Anterior chamber** | | |
| Ptosis | 1.08% | Deep | | 84.36% |
| Chalasis | 2.16% | Shallow | | 15.64% |
| Entropion | 0.00% | Cell | | 0.00% |
| MGD | 83.76% | Flare | | 0.00% |
| **Eyelashes** | | **Pupil** | | |
| Entropion | 0.00% | Round | | 97.83% |
| Trichiasis | 0.35% | Not round | | 2.17% |
| **Conjunctiva** | | Damaged | | 1.08% |
| Hyperemia | 14.08% | PE | | 4.69% |
| Papilla | 0.00% | **Lens** | | |
| Follicle | 2.67% | IOL | | 48.33% |
| Chalasis | 23.44% | NS 0 | | 0.00% |
| Pterygium | 6.74% | NS 1 | | 0.74% |
| **Cornea** | | NS 2 | | 22.30% |
| Opacity | 10.99% | NS 3 | | 20.82% |
| KP | 0.00% | NS 4 | | 5.95% |
| Bank keratopathy | 1.06% | NS 5 | | 1.86% |

**Fig 3. Characteristics and prevalence of anterior segment findings in the study population.** The average age was 85.69 ± 12.11 years, with a male-to-female ratio of 55:92. The most common eyelid abnormality was meibomian gland dysfunction. Other findings included conjunctival chalasis, and corneal opacity. Cataracts were present in all patients, with 48.33% having an intraocular lens.

## Severity distribution in lens findings

Lens findings were observable in 269 out of 294 eyes (91.50%). Of these, 48.33% were pseudophakic eyes (with IOLs), and no aphakic eyes were identified. The remaining 51.67% were evaluated as phakic eyes. The nuclear sclerosis classification of cataracts in the phakic eyes,

93 years of age, Male
Left Eye, Cataract

88 years of age, Female
Right Eye, Pterygium

96 years of age, Male
Right Eye, Bullous keratopathy

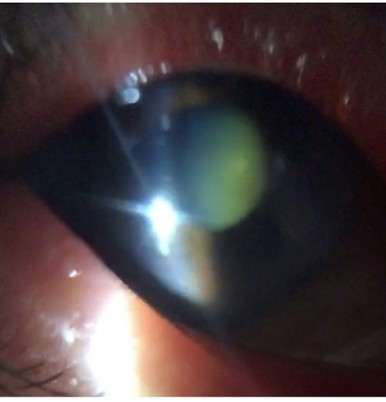 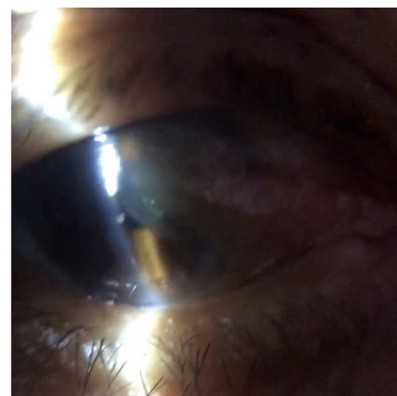 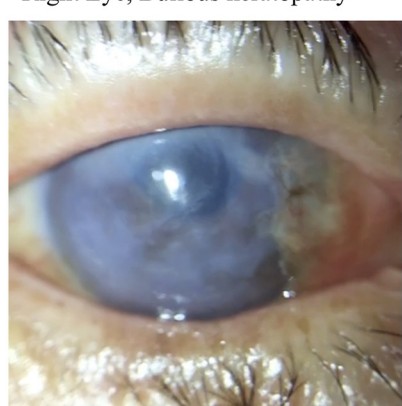

**Fig 4. Representative cases.** Slit-lamp photographs of anterior segment findings. Left: 93-year-old male with cataract in the left eye. Center: 88-year-old female with pterygium in the right eye. Right: 96-year-old male with bullous keratopathy in the right eye.

based on the Emery-Little classification, was as follows: 0.00% for NS0, 1.44% for NS1, 43.17% for NS2, 40.29% for NS3, 11.51% for NS4, and 3.60% for NS5 (Fig 5). Among the phakic eyes, 33.63% were found to have a shallow anterior chamber (Fig 5). A comparison of lens and anterior chamber findings across different age groups revealed that the incidence of higher NS values (indicating stronger nuclear sclerosis) increased with age. Additionally, the prevalence of shallow anterior chambers also increased with advancing age (Fig 5). Furthermore, a comparison between phakic and pseudophakic eyes indicated that the proportion of pseudophakic eyes increased in patients older than 90 years of age (Fig 5).

## Discussion

This cross-sectional study, conducted on Miyako Island, aimed to investigate the prevalence and severity of anterior segment diseases in patients receiving home-based care. The anterior segment of the eyes were imaged using a portable slit-lamp microscope by primary care physicians, and the findings were remotely evaluated by the ophthalmologists. All patients required home visits due to a decline in activities of daily living. The average age of the study subjects was comparable to the average life expectancy in Japan (Male: 81.1 years of age, Female: 87.1 years of age), and the cohort was predominantly female [18]. In this study, non-specialists used the SEC as a portable slit-lamp microscope, and specialists diagnosed the images remotely. Telemedicine using the SEC has been effectively utilized in other remote islands and for consultations between specialists, suggesting a high accuracy of doctor-to-doctor telemedicine [13].

A significant finding in this study was the extremely high prevalence of MGDs. The prevalence of MGD is typically reported to range from 11% to 63%, but the results in this study

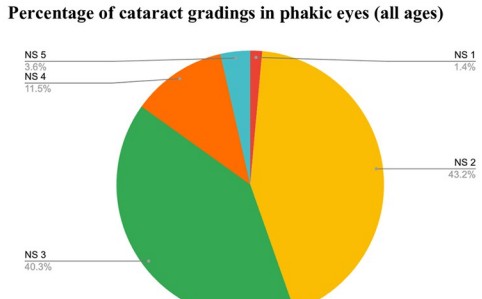

Percentage of cataract gradings in phakic eyes (all ages)

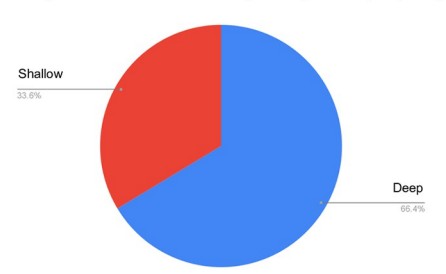

Percentage of anterior chamber depth in phakic eyes (all ages)

| All ages | | 50 years of age or under | | 51-60 years of age | | 61-70 years of age | |
|---|---|---|---|---|---|---|---|
| NS 0 | 0.00% | NS 0 | 0.00% | NS 0 | 0.00% | NS 0 | 0.00% |
| NS 1 | 1.44% | NS 1 | 0.00% | NS 1 | 33.33% | NS 1 | 0.00% |
| NS 2 | 43.17% | NS 2 | 100.00% | NS 2 | 33.33% | NS 2 | 80.95% |
| NS 3 | 40.29% | NS 3 | 0.00% | NS 3 | 16.67% | NS 3 | 19.05% |
| NS 4 | 11.51% | NS 4 | 0.00% | NS 4 | 16.67% | NS 4 | 0.00% |
| NS 5 | 3.60% | NS 5 | 0.00% | NS 5 | 0.00% | NS 5 | 0.00% |
| Deep | 66.37% | Deep | 100.00% | Deep | 66.67% | Deep | 78.95% |
| Shallow | 33.63% | Shallow | 0.00% | Shallow | 33.33% | Shallow | 21.05% |
| Phakia | 51.67% | Phakia | 66.67% | Phakia | 66.67% | Phakia | 90.48% |
| Psudophakia | 48.33% | Psudophakia | 33.33% | Psudophakia | 33.33% | Psudophakia | 9.52% |

| 71-80 years of age | | 81-90 years of age | | 91-100 years of age | | 101 years of age or older | |
|---|---|---|---|---|---|---|---|
| NS 0 | 0.00% | NS 0 | 0.00% | NS 0 | 0.00% | NS 0 | 0.00% |
| NS 1 | 0.00% | NS 1 | 0.00% | NS 1 | 0.00% | NS 1 | 0.00% |
| NS 2 | 45.45% | NS 2 | 47.06% | NS 2 | 21.74% | NS 2 | 0.00% |
| NS 3 | 36.36% | NS 3 | 37.25% | NS 3 | 60.87% | NS 3 | 0.00% |
| NS 4 | 18.18% | NS 4 | 11.76% | NS 4 | 13.04% | NS 4 | 50.00% |
| NS 5 | 0.00% | NS 5 | 3.92% | NS 5 | 4.35% | NS 5 | 50.00% |
| Deep | 75.00% | Deep | 72.55% | Deep | 69.57% | Deep | 50.00% |
| Shallow | 25.00% | Shallow | 27.45% | Shallow | 30.43% | Shallow | 50.00% |
| Phakia | 54.55% | Phakia | 52.58% | Phakia | 44.23% | Phakia | 20.00% |
| Psudophakia | 45.45% | Psudophakia | 47.42% | Psudophakia | 55.77% | Psudophakia | 80.00% |

**Fig 5. Prevalence of the lens and anterior chamber findings.** Pseudophakic eyes accounted for 48.33%, and 51.67% were phakic. Nuclear sclerosis (NS) grades increased with age, with NS2 and NS3 being the most common. Shallow anterior chambers were present in 33.63% of phakic eyes, and the proportion of pseudophakic eyes increased in patients over 90 years old.

exceeded these values [19]. Previous studies have noted that MGD increases or worsens with age. Aging is known to contribute to the pathogenesis of hyposecretory MGD, and recent MGD practice guidelines identify aging and menopause as risk factors [20]. The high prevalence of MGD in this study is understandable given the predominantly elderly and female cohort. Conjunctival findings revealed conjunctival hyperemia in approximately 16.6% of patients and conjunctival chalasis in 23.4%. Conjunctival chalasis, seen in many elderly patients, is characterized by the separation of the conjunctiva from the sclera, possibly caused by dilated subconjunctival lymph vessels or effusion. Dilated lymphatic vessels have been identified in 90% of such cases [21]. Reports on the prevalence of conjunctival hyperemia are limited due to its various causes, including bacterial or viral infections, allergies, and autoimmune conditions. Allergic conjunctival disease, which has a high prevalence of 48.7%, may also be influenced by seasonal and regional factors [22]. The prevalence of pterygium in this study was 6.74% (Fig 3). Ultraviolet light (UV) is considered a major cause of pterygium [23]. The Kume-jima Island Study reported a pterygium prevalence of 30.8% [24], while a Taiwanese study reported a prevalence of 2.14% and 3.48% in those over 40 years old [25]. The prevalence observed in this study likely falls between the results of these previous studies due to the geographic location of Miyakojima Island, which is situated between the two regions under comparison. Corneal opacity was present in 10.99% of cases, comparable to the 7.96% prevalence reported in a Taiwanese study of individuals aged 65 or older [23].

A notable finding was the high prevalence of cataracts, expected given the average age of over 85 years. However, more than half of the cases had not undergone cataract surgery, with many exhibiting moderate to severe cataracts.

A comparison of untreated cataracts in developing countries reveals significant disparities in cataract surgery coverage. In Ghana, 55% of the population aged 60–69 has undergone cataract surgery [26]. However, in our study, only 9.52% of individuals within the same age group received the surgery (Fig 5). Cataract surgery coverage varies significantly across regions, with Western Europe and North America achieving coverage rates ten times higher than those in Africa [27]. This regional variability complicates direct comparisons. In contrast, data from developed nations such as Australia show that 30–45% of individuals aged 79–84 have already undergone cataract surgery [28, 29]. Similarly, in the United Kingdom, 42.3% of the population aged 80 and above has undergone cataract surgery [30]. In the current study, the proportion of individuals who had not received cataract surgery was 1.2 to 1.5 times higher, likely due to increased rates of bedridden status and reduced activities of daily living (Fig 5).

The high prevalence is attributed to both age and Miyako Island's location at 24 degrees north latitude, resulting in higher UV exposure. UV exposure has been implicated in the development and progression of nuclear cataracts [31]. Currently, more than 1 million cataract surgeries are performed annually in Japan due to advancements in minimally invasive techniques [32]. However, the subjects in this study, living on a remote island and requiring home visits, may have limited access to ophthalmic care, including surgery [33]. About one-third of the patients with lenses had shallow anterior chamber depths, a known risk for angle closure, potentially leading to acute glaucoma attacks [34]. Previous reports indicate approximately 1.3% of Japanese over 40 years have angle closure. Although anterior chamber depth was not accurately measured in this study, a higher frequency of shallow anterior chambers was identified compared to previous reports. Risk factors for angle closure include abnormal lens position, increased lens thickness due to cataract progression, and overmatured cataracts [35]. This study's limitations include its retrospective nature and cross-sectional observational design, which did not examine treatments implemented for diagnosed cases or collaboration with ophthalmology specialists. Further research is needed on collaborative treatment with local ophthalmologists. Given that ophthalmology is a diagnostic imaging-focused specialty,

similar collaborative efforts between local non-ophthalmologists and remote ophthalmologists can be conducted in various regions [36, 37]. The study highlights the potential for telemedicine to bridge the gap in ophthalmic care in remote areas. By utilizing portable imaging devices and remote specialist consultations, high-quality care can be delivered to populations that are otherwise underserved due to geographic and resource limitations. This model could be replicated in other regions, improving access to ophthalmic care for home-bound and elderly populations. This study provides valuable insights into the prevalence and severity of anterior segment diseases among elderly home-bound patients in Miyako Island. The high prevalence of MGD, conjunctival chalasis, pterygium, corneal opacity, and cataracts underscores the need for enhanced ophthalmic care in this demographic.

The findings advocate for the integration of telemedicine and portable diagnostic tools in routine ophthalmic practice to ensure comprehensive care for patients with limited access to traditional healthcare facilities.

## Conclusions

This study identified a significant presence of untreated anterior segment diseases, primarily cataracts, among patients receiving home-visit care on Miyako Island. These patients, residing on remote islands, are at risk of missing essential ophthalmologic care opportunities. Therefore, there is a critical need to establish a robust medical care support system to ensure these patients receive adequate ophthalmologic evaluation and treatment, encompassing both public and private sectors. Given the limitations of medical resources, the integration of information and communications technology -based telemedicine and diagnostic support systems, such as the SEC used in this study, should be strongly considered to enhance the accessibility and quality of ophthalmologic care for this vulnerable population.

## Author Contributions

**Conceptualization:** Eisuke Shimizu, Yusuke Shimizu.

**Data curation:** Kazuhiro Hisajima, Shintaro Nakayama, Hiroki Nishimura, Rohan Jeetendra Khemlani.

**Formal analysis:** Eisuke Shimizu, Shintaro Nakayama.

**Funding acquisition:** Eisuke Shimizu.

**Investigation:** Kazuhiro Hisajima, Hiroki Nishimura, Rohan Jeetendra Khemlani, Yusuke Shimizu.

**Methodology:** Eisuke Shimizu, Shintaro Nakayama, Ryota Yokoiwa, Yusuke Shimizu, Keigo Yasukawa.

**Project administration:** Eisuke Shimizu, Ryota Yokoiwa.

**Resources:** Shintaro Nakayama, Hiroki Nishimura, Rohan Jeetendra Khemlani, Ryota Yokoiwa, Keigo Yasukawa.

**Software:** Shintaro Nakayama.

**Supervision:** Masato Kishimoto, Keigo Yasukawa.

**Validation:** Kazuhiro Hisajima, Shintaro Nakayama, Hiroki Nishimura, Rohan Jeetendra Khemlani, Ryota Yokoiwa, Yusuke Shimizu, Masato Kishimoto, Keigo Yasukawa.

**Visualization:** Ryota Yokoiwa, Masato Kishimoto.

**Writing – original draft:** Eisuke Shimizu.

**Writing – review & editing:** Kazuhiro Hisajima, Shintaro Nakayama, Hiroki Nishimura, Rohan Jeetendra Khemlani, Ryota Yokoiwa, Yusuke Shimizu, Masato Kishimoto, Keigo Yasukawa.

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
