## [Decision Letter · Decision Letter 0]

1 Sep 2024

PONE-D-24-24324Epidemiological Survey of Anterior Segment Diseases in Japanese Isolated Island Using a Portable Slit-Lamp Device in Home-Based Cases in Miyako IslandPLOS ONE

Dear Dr. Shimizu,

Thank you for submitting your manuscript to PLOS ONE. After careful consideration, we feel that it has merit but does not fully meet PLOS ONE’s publication criteria as it currently stands. Therefore, we invite you to submit a revised version of the manuscript that addresses the points raised during the review process. Please submit your revised manuscript  by  Oct 16 2024 11:59PM. If you will need more time than this to complete your revisions, please reply to this message or contact the journal office at plosone@plos.org. Please include the following items when submitting your revised manuscript:A rebuttal letter that responds to each point raised by the academic editor and reviewer(s). You should upload this letter as a separate file labeled 'Response to Reviewers'.A marked-up copy of your manuscript that highlights changes made to the original version. You should upload this as a separate file labeled 'Revised Manuscript with Track Changes'.An unmarked version of your revised paper without tracked changes. You should upload this as a separate file labeled 'Manuscript'.If applicable, we recommend that you deposit your laboratory protocols in protocols.io to enhance the reproducibility of your results. Protocols.io assigns your protocol its own identifier (DOI) so that it can be cited independently in the future. For instructions see: https://journals.plos.org/plosone/s/submission-guidelines#loc-laboratory-protocols. Additionally, PLOS ONE offers an option for publishing peer-reviewed Lab Protocol articles, which describe protocols hosted on protocols.io. Read more information on sharing protocols at https://plos.org/protocols?utm_medium=editorial-email&utm_source=authorletters&utm_campaign=protocols.

We look forward to receiving your revised manuscript.

Kind regards,

Jiro Kogo

Academic Editor

PLOS ONE

Journal Requirements:

2. Thank you for stating the following financial disclosure: This work was supported by the Japan Agency for Medical Research and Development, Uehara Memorial Foundation, Hitachi Global Foundation, Kondo Memorial Foundation, Eustylelab, Kowa Life Science Foundation, Keio University Global Research Institute, The Asahi Glass Foundation, The Okawa Foundation, Suzuken Memorial Foundation, Yakult bioscience Foundation and Daiwa Securities Foundation. No other funding statement to declare including company and patent.

3. Thank you for stating the following in the Competing Interests section: E.S. is a founder of OUI Inc. OUI Inc. has the patent for the Smart Eye Camera (Patent No. JP; 6627071, USA; 16/964822, EU; 19743494.7, China; 201980010174.7, India; 202017033428, VN; 1-2020-04893, and Africa; AP/P2020/012569. Patent pending EU; 2175926.2, US; 17/799043). There are no other relevant declarations relating to this patent. OUI Inc. did not have any role in the funding, study design, data collection and analysis, decision to publish, or preparation of the manuscript. The authors declare no competing interest associated with this manuscript.

Reviewers' comments:

Reviewer's Responses to Questions

**Comments to the Author**

1. Is the manuscript technically sound, and do the data support the conclusions?

Reviewer #1: Yes

Reviewer #2: Yes

2. Has the statistical analysis been performed appropriately and rigorously? 

Reviewer #1: N/A

Reviewer #2: Yes

3. Have the authors made all data underlying the findings in their manuscript fully available?

Reviewer #1: Yes

Reviewer #2: Yes

4. Is the manuscript presented in an intelligible fashion and written in standard English?

Reviewer #1: Yes

Reviewer #2: Yes

5. Review Comments to the Author

Reviewer #1: This study is an epidemiological investigation targeting an elderly population that faces significant barriers to accessing ophthalmic care and is consequently less likely to receive appropriate treatment. Given the isolated environment of the remote island and the limited number of home care patients, the sample size is considered appropriate when considering the average age of the patients.

The portable slit-lamp microscope developed by the authors can evaluate a variety of anterior segment diseases. This study implies that visiting physicians who are not ophthalmologists can obtain critical information about anterior segment diseases. Previous reports have shown to have significant potential for widespread application beyond Japan. This study and portable slit-lamp microscope have the potential to benefit rural regions. However, some clarification on the discussion within this paper may be needed.

The finding that many anterior segment diseases, particularly cataracts, remain untreated among patients on isolated islands is noteworthy. However, the current phrasing may need to establish whether the prevalence of untreated cataracts is indeed high. It might be more informative to compare the rate of previous cataract surgeries among patients over 80 years in this study with those in similar age groups in both developed and developing countries. Given that approximately half of the patients in this study underwent cataract surgery, it is possible that factors other than access to health care alone are involved. If the aim is to emphasize the high prevalence of cataracts in this population, referencing reports from comparable cohorts could provide a more compelling argument.

The discussion suggests that the prevalence of pterygium is not high in this study compared to other reports. The authors show the reason could be the subjects' predominantly home-bound lifestyle, which reduces exposure to ultraviolet (UV). However, this contrasts with the explanation for the high prevalence of cataracts, which is attributed to increased UV exposure among the subjects. This hypothesis regarding UV exposure may create a potential inconsistency in the consideration. Because cataracts and pterygium have multifactorial etiologies, making definitive statements is challenging. If the intent is to emphasize the high prevalence of cataracts, it might be advisable to refrain from attributing the prevalence of pterygium to the home-bound lifestyle.

Reviewer #2: The manuscript by Shimizu et al., titled "Epidemiological Survey of Anterior Segment Diseases in Japanese Isolated Island Using a Portable Slit-Lamp Device in Home-Based Cases in Miyako Island" is a report on epidemiological survey of anterior segment diseases amongst people living on Miyako island, which is a Japanese isolated island using a portable slit-lamp device. The study was a home-based survey. The report will be intereting to colleagues specialising in ophthalmology and public health. The manuscript is well-written and analysis properly done.

6. PLOS authors have the option to publish the peer review history of their article (what does this mean?). If published, this will include your full peer review and any attached files.

Reviewer #1: No

Reviewer #2: **Yes: **Professor Emmanuel oluwadare Balogun

---

## [Author Response · Author response to Decision Letter 0]

29 Sep 2024

Response to the Reviewer

26th September, 2024

Dr. Eisuke Shimizu

Reviewer #1:

Comment:

"The study finds that many anterior segment diseases, particularly cataracts, remain untreated among patients on isolated islands. However, the current phrasing may need to establish whether the prevalence of untreated cataracts is indeed high. It might be more informative to compare the rate of previous cataract surgeries among patients over 80 years in this study with those in similar age groups in both developed and developing countries. Given that approximately half of the patients in this study underwent cataract surgery, it is possible that factors other than access to healthcare alone are involved."

Response:

Thank you for this insightful suggestion. We agree that a direct comparison of cataract surgery rates in similar age groups across different regions could strengthen our argument. In the revised manuscript, we have included comparisons with data from developed countries as well as from developing regions where access to ophthalmic care is limited. We have modified the discussion to incorporate these multifactorial considerations.

LINE 248

A comparison of untreated cataracts in developing countries reveals significant disparities in cataract surgery coverage. In Ghana, 55% of the population aged 60–69 has undergone cataract surgery [26]. However, in our study, only 9.52% of individuals within the same age group received the surgery (Fig 5). Cataract surgery coverage varies significantly across regions, with Western Europe and North America achieving coverage rates ten times higher than those in Africa [27]. This regional variability complicates direct comparisons. In contrast, data from developed nations such as Australia show that 30-45% of individuals aged 79–84 have already undergone cataract surgery [28, 29]. Similarly, in the United Kingdom, 42.3% of the population aged 80 and above has undergone cataract surgery [30]. In the current study, the proportion of individuals who had not received cataract surgery was 1.2 to 1.5 times higher, likely due to increased rates of bedridden status and reduced activities of daily living (Fig 5).

Comment:

"The discussion suggests that the prevalence of pterygium is not high in this study compared to other reports. The authors show the reason could be the subjects' predominantly home-bound lifestyle, which reduces exposure to ultraviolet (UV). However, this contrasts with the explanation for the high prevalence of cataracts, which is attributed to increased UV exposure among the subjects. This hypothesis regarding UV exposure may create a potential inconsistency."

Response: 

We appreciate your observation regarding the apparent inconsistency in the explanation of UV exposure's role in cataracts and pterygium prevalence. To address this, we have revised the manuscript to clarify that while UV exposure is a significant risk factor for both conditions, the multifactorial nature of their etiology means that different factors might dominate in different settings. This updated explanation has been integrated into the discussion to present a more balanced and consistent interpretation.

LINE 237

The prevalence observed in this study likely falls between the results of these previous studies due to the geographic location of Miyakojima Island, which is situated between the two regions under comparison.

Reviewer #2:

Comment:

"The manuscript by Shimizu et al., titled 'Epidemiological Survey of Anterior Segment Diseases in Japanese Isolated Island Using a Portable Slit-Lamp Device in Home-Based Cases in Miyako Island,' is a report on an epidemiological survey of anterior segment diseases amongst people living on Miyako island. The report will be interesting to colleagues specializing in ophthalmology and public health. The manuscript is well-written and the analysis properly done."

Response:

Thank you for your positive feedback. We are grateful for your comment and we continued to refine the manuscript for clarity and precision as we finalize the revisions.

LINE 237

The prevalence observed in this study likely falls between the results of these previous studies due to the geographic location of Miyakojima Island, which is situated between the two regions under comparison.

LINE 248

A comparison of untreated cataracts in developing countries reveals significant disparities in cataract surgery coverage. In Ghana, 55% of the population aged 60–69 has undergone cataract surgery [26]. However, in our study, only 9.52% of individuals within the same age group received the surgery (Fig 5). Cataract surgery coverage varies significantly across regions, with Western Europe and North America achieving coverage rates ten times higher than those in Africa [27]. This regional variability complicates direct comparisons. In contrast, data from developed nations such as Australia show that 30-45% of individuals aged 79–84 have already undergone cataract surgery [28, 29]. Similarly, in the United Kingdom, 42.3% of the population aged 80 and above has undergone cataract surgery [30]. In the current study, the proportion of individuals who had not received cataract surgery was 1.2 to 1.5 times higher, likely due to increased rates of bedridden status and reduced activities of daily living (Fig 5).

---

## [Editor Report · Decision Letter 1]

4 Oct 2024

Epidemiological Survey of Anterior Segment Diseases in Japanese Isolated Island Using a Portable Slit-Lamp Device in Home-Based Cases in Miyako Island

PONE-D-24-24324R1

Dear Prof. Shimizu

We’re pleased to inform you that your manuscript has been judged scientifically suitable for publication and will be formally accepted for publication once it meets all outstanding technical requirements.

Kind regards,

Jiro Kogo

Academic Editor

PLOS ONE

---

## [Editor Report · Acceptance letter]

13 Nov 2024

PONE-D-24-24324R1 

PLOS ONE

Dear Dr. Shimizu, 

I'm pleased to inform you that your manuscript has been deemed suitable for publication in PLOS ONE. Congratulations! Your manuscript is now being handed over to our production team.

Kind regards, 

on behalf of

Prof. Jiro Kogo 

Academic Editor

PLOS ONE